# qRobot: A Quantum Computing Approach in Mobile Robot Order Picking and Batching Problem Solver Optimization

**Parfait Atchade-Adelomou** [1,2,*]![ID], **Guillermo Alonso-Linaje** [3,*], **Jordi Albo-Canals** [2,*] **and Daniel Casado-Fauli** [1,*]

1   Research Group on Data Science for the Digital Society, La Salle, Universitat Ramon Llull, Carrer de Sant Joan de La Salle, 42, 08022 Barcelona, Spain
2   Lighthouse Disruptive Innovation Group, LLC., 7 Broadway Terrace, Apt. 1, Cambridge, MA 02139, USA
3   Facultad de Ciencias, Campus Miguel Delibes, Universidad de Valladolid, C/Plaza de Santa Cruz, 8, 47002 Valladolid, Spain
*   Correspondence: parfait.atchade@salle.url.edu or parfait.atchade@lighthouse-dig.com (P.A.-A.); guillermo.alonso.alonso-linaje@alumnos.uva.es (G.A.-L.); jordi.albo@lighthouse-dig.com (J.A.-C.); daniel.casado@salle.url.edu (D.C.-F.)

**Abstract:** This article aims to bring quantum computing to robotics. A quantum algorithm is developed to minimize the distance traveled in warehouses and distribution centers where order picking is applied. For this, a proof of concept is proposed through a Raspberry Pi 4, generating a quantum combinatorial optimization algorithm that saves the distance travelled and the batch of orders to be made. In case of computational need, the robot will be able to parallelize part of the operations in hybrid computing (quantum + classical), accessing CPUs and QPUs distributed in a public or private cloud. We developed a stable environment (ARM64) inside the robot (Raspberry) to run gradient operations and other quantum algorithms on IBMQ, Amazon Braket (D-Wave), and Pennylane locally or remotely. The proof of concept, when run in the above stated quantum environments, showed the execution time of our algorithm with different public access simulators on the market, computational results of our picking and batching algorithm, and analyze the quantum real-time execution. Our findings are that the behavior of the Amazon Braket D-Wave is better than Gate-based Quantum Computing over 20 qubits, and that AWS-Braket has better time performance than Qiskit or Pennylane.

**Keywords:** quantum computing; machine learning; picking problem; batching problem; quantum robotics; Raspberry PI4; docplex

## 1. Introduction

From DHL, Gartner, and others [1–3], we know that the first wave of automation using smart robotics has reached the logistics industry. Driven by rapid technological advancements and increased affordability, robotic solutions (software and hardware) are forcibly entering labor logistics, supporting flawless processes and boosting productivity. Robots, especially mobile, will adopt more roles in the supply chain, helping workers with storage, transportation and little by little, they will expand their service. In fact, in some countries, there are already robotic delivery services [4].

We are already living an exponential increment of mail-order shopping, online shopping and supply chain systems, requiring large-scale logistic centers. Almost everyone can order products remotely, and the logistic center increases its functionalities, including keeping and shipping products. While there was a tendency to increase the adoption of automated systems based on robots powered by AI to increase efficiency [5–7], COVID-19 introduced the concept of touch-less online shopping that reduces the risk of infections. Smart Warehouses are the epicenter of the cost-efficiency of any e-commerce company [8].

The emerging field of hybrid (quantum-classical) algorithms joins CPU, and QPU [9] to speed up specific calculations within a classical algorithm. This allows for shorter

quantum runs that are less susceptible to the cumulative effects of noise and work well in current devices.

Recently, the scientific community has begun researching the real implementation of quantum computing algorithms in mobile platforms because implementations are not here yet [10].

In this article, we demonstrate that we can implement this system in a well-known, widely used in robotics fields, computer system, like raspberry-pi, exploring the performance of a quantum picking and batching model. A hybrid system is proposed to effectively replace the current ones and open the doors to quantum computing in robotics. In addition, we are analyzing the results obtained with different public access simulators on the market: IBMQ, Amazon Braket (D-Wave), and Pennylane. As far as the authors know, this is the first time this type of implementation has been done.

After Section 1, the document is organized as follows; Section 2 shows previous work on both assembly techniques and approaches to picking and batch management systems; then, Section 3 presents the quantum fundamentals needed from this era to solve this problem; next, the implementation of the proposed strategy and the creation of the qRobot performed in Section 4 are explained; to continue, Section 5, which shows the results of our experimental analysis, and Section 6, in which some open problems are summarized, compared, and presented; and, finally, Section 7 concludes the previous results and describes the future work.

## 2. Work Context

According to Reference [11–13], supply chains, warehouses, and distribution centers occupy a very important position when storing and serving customer demand. Today, in order to be competitive within this sector, Logistics 4.0 has been created, which is known as the set of artificial intelligence technologies and techniques that seek the efficiency of the movements of materials and products in a factory or warehouse. Better time management helps logistics companies find and locate a material, reduce fatigue and possible workplace accidents, and spend less time documenting items.

Many works of literature highlight these factors as the main ones where the loss of time and resources in a process require an urgent solution, and, precisely, it is technologies, such as Artificial Intelligence and the Internet of Things (IoT), which today allow us to optimize them [14–16].

Only in the last decade, researchers have focused on addressing the multiple order picking planning problems. The study of the efficiency of a Warehouse can be addressed based on multiple parameters. According to Reference [17], there are three key considerations: (1) Performance Measure (time, cost, productivity, and service), (2) How we model the warehouse (Analytical model, Mathematical Model, or Simulation), and (3) the combination of factors (storage location assignment, routing, order batching, or other order picking planning problems).

Based on data from Reference [17], we can see the percentage of relevance of the considered order picking planning problems based on the percentage of papers that are related to such challenges.

As we can see in Figure 1, Picking and Batching are the top priorities based on the research contributions.

Order preparation (picking) is one of the most frequent and costly operations in labor [11,12] since it is responsible for recovering the items required by the orders of customer orders (could also be supplied, but, in this article, we focus exclusively on sales orders), and to create the batches, grouping several orders of orders in a picking list to collect all the batch demands in a single warehouse tour. In this last part of order preparation, our quantum algorithm comes into action to optimize the routes traveled to achieve efficient picking.

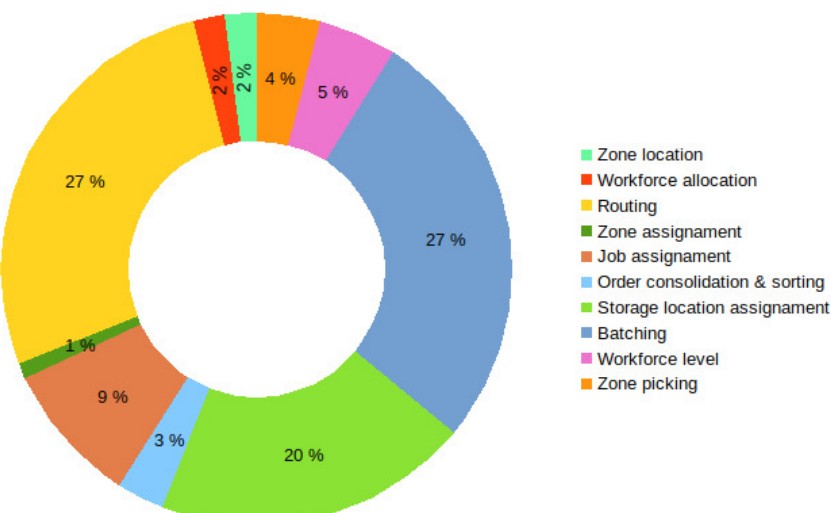

**Figure 1.** Distribution of considered order picking problems based on percentage of publications.

There are many techniques and strategies for solving the picking problem. The most striking are "The selected techniques for evaluation include A *" [18], "Potential Fields (PF)", "Rapidly-Exploring Random Trees * (RRT *)" [19–21], and "Variations of the Fast-Marching Method (FMM)" [22]. Other strategies have explored using the TSP and the VRP as algorithms to solve the picking problem. In this case, if the number of order orders per lot is greater than two [23], picking becomes an NP-Hard problem in which the number of possible lots and binary variables increase exponentially with the number of purchase orders [23]. From there, several heuristic techniques, methods and algorithms (for example, genetic) were created to relax these difficulties [11,13,16,24–26]. However, and as mentioned above, depending on the volume of data, the computational cost of the algorithm becomes intractable for classical computing.

The latter leads us to explore new approaches to the large-scale picking problem, and one of the approaches to take into account to solve this task is quantum computing [6]. Quantum computing could help us change the degree of complexity of the problem, enhanced by its high computing power. Among the great fields where quantum computing is called to stand out is constraint satisfaction problems (CSP) [27]. One of the useful algorithms in this field is Quadratic Unconstrained Binary Optimization (QUBO) problems [28].

From Alan Turing [29] to Richard Feyman's idea of considering the simulation of systems in quantum mechanics by other quantum systems [30], interest in creating new ways of solving them has grown dramatically. This, together with the consequences of the well-known Moore's Law, gave way to the idea of building quantum computers. Over the past decades, before demonstrating the superiority of quantum computing, David Deutsch published his work [31] in which he proposed how a universal quantum computer could be. Years later, the worth of these new computers has been demonstrated to solve some specific problems, such as factoring prime numbers using Shor's [32] algorithm or searching in disordered sets with Grover's [33] algorithm, although all this limited to the number of qubits available. We are currently in the NISQ era [34], in which we have computers between 50 and 100 qubits (Gate-Based Quantum Computer), opening the way to the emerging field of hybrid quantum-classical computing. Within this, different algorithms have been developed, such as "VQE" [35], "QAOA" [36], or "Quantum Machine Learning (QML)" [37–42], which we will focus on with this article.

There are two dominant techniques for quantum computing: Continuous-Time Quantum Computing [43] used by D-Wave, in which the problem to solve is mapped in quantum hamiltonians and the natural dynamics of physical systems; and the Gate-based Quantum Computing [44,45] led by IBM, in which the computation is made through a series

of discrete gate operations. Reference [43] argues how Quantum Walk (QW), Quantum Annealing (QA), and Adiabatic Quantum Computing (AQC) are related. QW and AQC are pure quantum evolutions (unitary), while QA involves external cooling.

Adiabatic Quantum Computing, proposed by Farhi [46,47], is based on the adiabatic theorem [44] and was the first quantum computing technique.

Quantum Annealing, based on the adiabatic quantum computing paradigm, was initially introduced by Kadowaki and Nishimori [48]. Since its proposal, the QA technique was essential for solving combinatorial optimization problems. This technique tries to solve problems similar to how optimization problems are solved using the classical simulated annealing [44]. It is known that from a multivariate function formed from an energy landscape, the ground state corresponds to the optimal solution of the problem. Therefore, the quality assurance process should be repeated until the optimal solution to the problem is found. The most significant advantage of quantum annealing is its high degree of parallelism over classical code execution. Because it analyzes all possible inputs in parallel to find the optimal solution, this is very useful when we want to reduce the complexity of the NP-complete problems.

QA has confirmed its ability to solve a broad range of combinatorial optimization problems, and also in other fields, such as quantum chemistry [44], bioinformatics [44], and routing [49], to cite a few.

We can categorize combinatorial optimization problems into several groups, where there is a need for adequate techniques for solving such problems. One of the standardized optimization problems is the aforementioned QUBO [44,48,50].

QUBO, as NP-hard, refers to a pattern matching technique that, among other applications, can be used in machine learning and optimization and which involves minimizing a quadratic polynomial on binary variables [44]. QUBO has demonstrated its potential in solving some standard combinatorial optimization problems, such as the coloring of graphics, workshop planning, vehicle routing and programming, neural networks, the partition problem, 3-SAT, and machine learning, where the parameters of the problem can be expressed as Boolean variables [44,48,50], only to remember that Adiabatic quantum annealing techniques are also used to solve multi-objective optimization problems [51]. The QUBO formulation is suitable for running a D-Wave architecture; nevertheless, QUBO can be mapped on the Ising model [44].

Advances in quantum computing offer a way forward for efficient solutions to many cases of substantial eigenvalue problems unsolvable in a traditional way [52]. Quantum approaches to finding eigenvalues previously relied on the Quantum Phase Estimation (QPE) algorithm. The QPE is one of the essential subroutines in quantum computation. It serves as a central building block for many quantum algorithms and offers exponential acceleration compared to classical methods, and it requires several quantum operations $O\left(\frac{1}{p}\right)$ to obtain an estimate with precision $p$ [52,53].

Variational Quantum Eigensolver (VQE), proposed by Peruzzo [52], based on the variational principle and form, estimates the ground state energy of the Hamiltonian of the problem [54]. The VQE is a hybrid quantum/classical algorithm originally proposed to approximate the ground state of a quantum system (the state attaining the minimum energy). Quantum Approximate optimization Algorithms (QAOA), based on the principles of adiabatic quantum computation [44,53], is used to solve QUBO problems. Farhi and Harrow showed the advantages of QAOA compared to classical approaches [46,47]. While [55] debated the problems of constrained polynomial optimization using adiabatic quantum computation methods, other scientists, such as Vyskocil and Djidjev [56], worked on how to apply restrictions in QUBO systems to avoid the use of large numbers of the coefficients. This resulted in more qubits from the use of quadratic penalties, they proposed a new combinatorial design which involved solving problems of linear programming of mixed integers to adapt applications restitution. Anuradha Mahasinghe, Richard Hua, Michael J. Dinneen, and Rajni Goyal [57] investigated and solved the Hamiltonian cycle problem in computational frameworks, such as quantum circuits, quantum walks, and adiabatic

quantum computing. All of these advances in quantum computing have been applied to routing and scheduling techniques. Reference [58] contributed an expansive vision and discussions on Ising formulations for various NP-complete and NP-hard optimization problems, emphasizing using as the fewest possible as possible qubits. In the same way, there have been many works of literature on the VRP [59] and its variants.

Amazon Braket [60] is a cloud-based (Figures 2 and 3), fully managed quantum computing service that helps researchers and developers get started into quantum world technology to accelerate research and discovery. Amazon Braket provides a development environment to explore, create, test, and run quantum algorithms, quantum circuit simulators, and different quantum hardware technologies.

We will take advantage of all these related works to define an appropriate strategy for our proposal in this NISQ era.

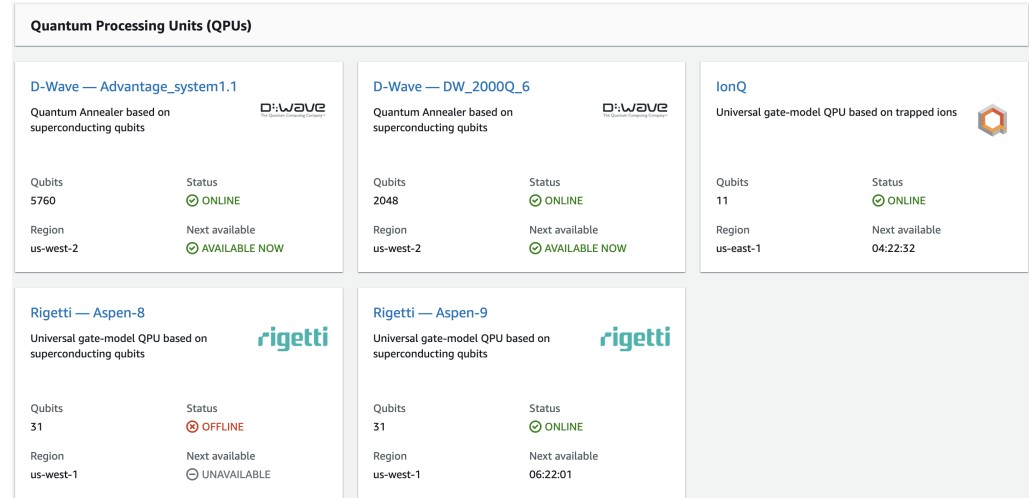

**Figure 2.** Technical specifications of the Quantum Hardware Technologies (Gate-based superconducting qubits, Gate-based ion traps, and Quantum annealing) available in Amazon Braket.

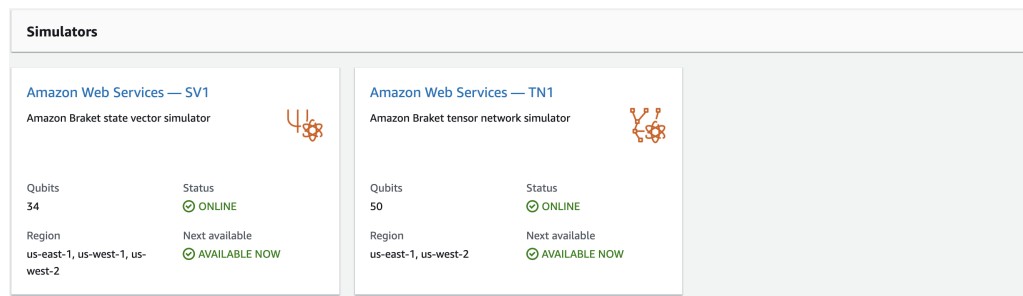

**Figure 3.** Technical Specifications of Quantum Simulators systems where we can see the state vector simulator (34 qubits) and tensor network simulator (50 qubits).

Studying and comparing different optimization methods of warehouse's challenge, like picking and batching, ref. [17] proposes three options: analytical models, simulation experiments, and mathematical programming. In our approach, we consider the latter. We use a set of mathematical expressions that describe the problem, represented by an objective mathematical function and constraints within the classical context and translate it to the quantum domain.

While reviewing state of the art research, Reference [61] was found. The integrated order routing and the batch problem is modeled in such systems as an extended multi-tank vehicle routing problem with network flow formulations of three indices and two commodities. Such a variable neighborhood search algorithm provides close to optimal solutions within a computational time acceptable for classical but not quantum computing.

This article intends to bring quantum computing to robotics by proposing an approach that combines the experience of classical robotics computing with the computation of complex and high-cost processes by quantum computing. We suggest preparing an environment to execute the quantum algorithms in the mobile and autonomous robot remotely and locally and design a quantum algorithm that helps the efficiency of the warehouse management.

## 3. Quantum Circuits in the NISQ Era

Quantum circuits are defined mathematically as actions in an initial quantum state. Quantum computing largely uses quantum states constructed from qubits, namely binary states represented by $|\psi\rangle = \alpha|0\rangle + \beta|1\rangle$. Its number of qubits $n$ commonly defines the states of a quantum circuit, and, normally, the initial state of the circuit $|\psi\rangle_0$ is the zero state $|0\rangle$. Mostly, a quantum circuit implements an internal unit operation $U$ in the initial state $|\psi\rangle_0$ to transform it into the final output state $|\psi\rangle_f$. This $U$ gate is normally fixed and is known for algorithms or problems. In contrast, other methods define its internal operation through a fixed structure, called Ansatz [62], and adjustable parameters $\theta$ [63]. Parameterized circuits are beneficial and have interesting properties in this quantum era, as they broadly define the definition of ML and offer the flexibility and viability of unit operations with arbitrary accuracy [37–39,64].

*Variational Quantum Eigensolver*

The Variational Quantum Eigensolver (VQE) [35] is a classical hybrid quantum algorithm that combines aspects of quantum mechanics with the classical algorithm (Figure 4). Its great contribution is to find approximate solutions to combinatorial problems. Its operation is based on mapping the combinatorial problems in a physics problem, i.e., about a problem that can be formulated in terms of a Hamiltonian Ising model. Therefore, identifying the solution to the combinatorial problem is linked to finding the ground state of this physics problem. Thus, the goal is to find the ground state of this Hamiltonian. The unknown eigenvectors are prepared by varying the experimental parameters and calculating the Rayleigh-Ritz ratio [65] in a classical minimization (Figure 5). At the end of the algorithm, the reconstruction of the eigenvector stored in the final set of experimental parameters that define the state will be done.

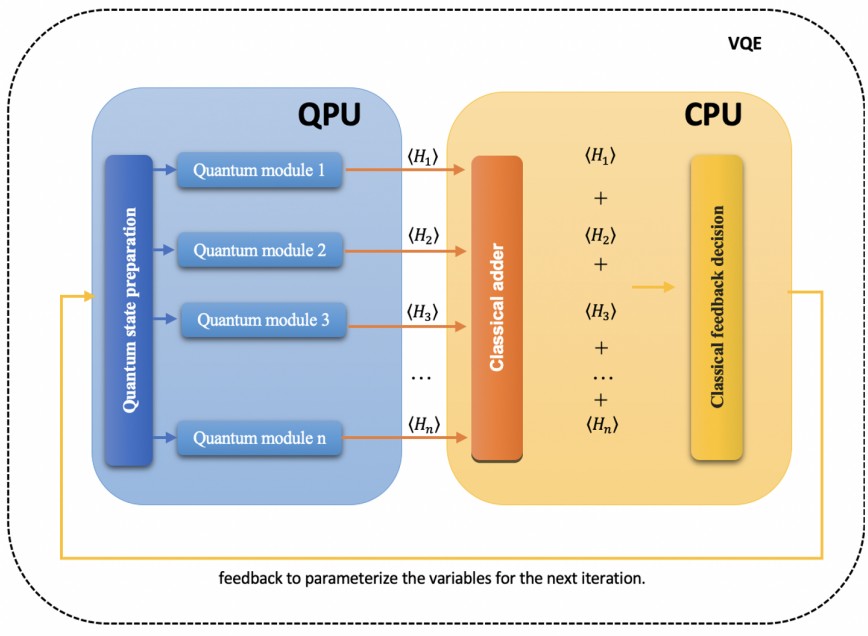

**Figure 4.** The Variational Quantum Eigensolver diagram.

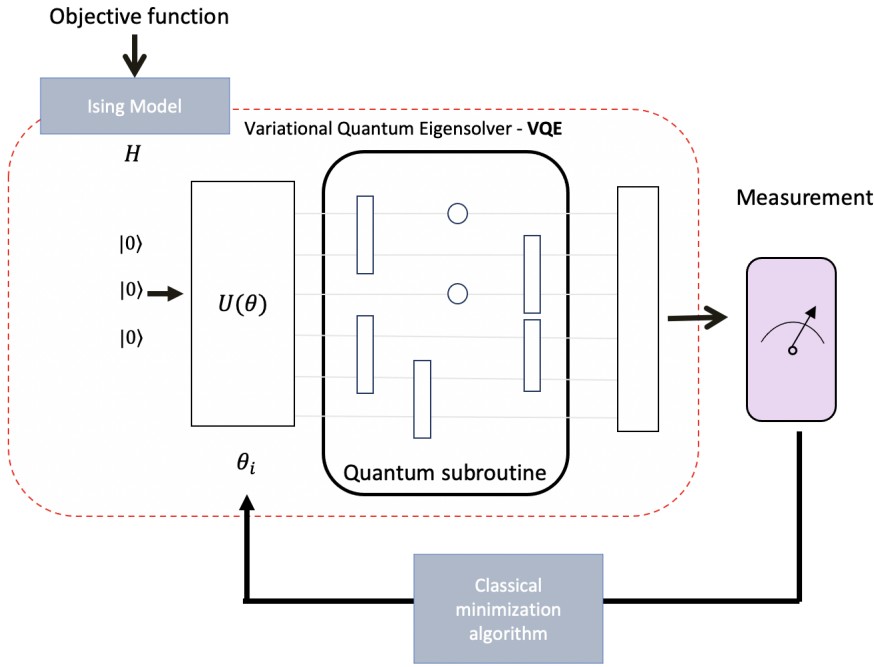

**Figure 5.** VQE working principle based on the quantum variational circuit.

From the variational principle, the following equation $\langle H \rangle_{\psi(\vec{\theta})} \geq \lambda_i$ can be reached out, with $\lambda_i$ as eigenvector, and $\langle H \rangle_{\psi(\vec{\theta})}$ as the expected value. In this way, the VQE finds (1) such an optimal choice of parameters $\vec{\theta}$, that the expected value is minimized and that a lower eigenvalue is located.

$$\langle H \rangle = \langle \psi(\theta) | H | \psi(\theta) \rangle. \tag{1}$$

We will use the VQE (Figure 5) to find the minima of our objective function translated to the Ising model.

## 4. Implementation

To carry out the implementation of our proof of concept (Figures 6 and A1), we must first prepare the programming environment. Considering that the core of our robot will be the Raspberry Pi 4 [66], the first thing to do is prepare it so that it can execute quantum algorithms with the guarantees required for the proposed application and especially for future operations on gradients. It is necessary to install an ARM64 operating system [67,68] with all the needed packages to run all the required environments to carry out this project. We took advantage of the work for Raspberry Pi Os Desktop (32-bit) on which the author describes how to install and run Qiskit—IBM's open-source quantum computing software framework [69]—on a Raspberry Pi to turn it into a quantum computing simulator and use it to access real IBM quantum computers. In our case, we do need ARM64 because we need to execute at least the TensorFlow's version 3.2.1. The tasks to convert the Raspberry Pi 4 in our "quantum computer" are in the Appendix A.

After setting up the environment, we will focus on designing and experimenting with the announced proof of concept.

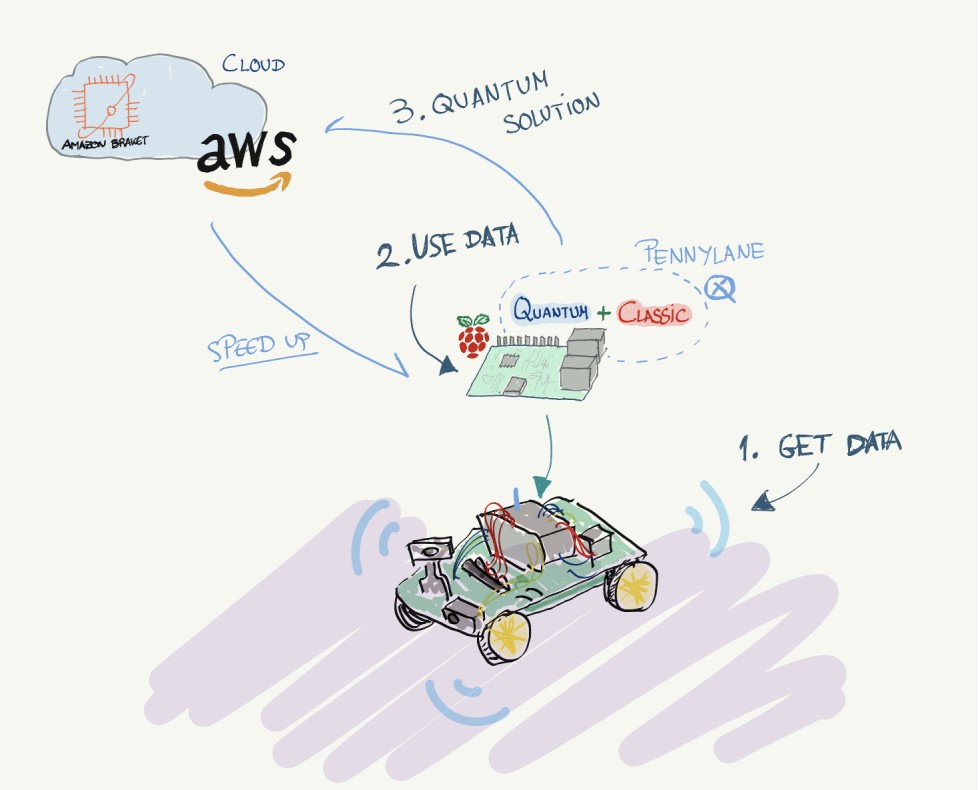

**Figure 6.** We propose a robot that prepares batches and increases the efficiency of picking in a warehouse, taking advantage of the classic Machine Learning experience and leveraging hybrid computing (classical + quantum) in the cloud and distributed. This robot uses the Optimal routing strategy to calculate the shortest route, regardless of the layout or location of the items.

*4.1. The Problem's Formulation*

In this formulation, we will seek to optimize the collection of the products and, later, we will make the batches.

To carry it out, we will consider the following assumptions:

1. The strategy we will follow is the picking routing problem to retrieve each lot which the total distance traveled to retrieve all the items in a lot will be calculated.
2. The warehouse configuration is given in Figures 7–9 .
3. For the orders of the storage positions, more than one picking robot can be used.
4. Movements in height are not considered.
5. Each product is stored in a single storage position, and only one product is stored in each storage position.
6. Each picking route begins and ends at the Depot.
7. The load capacity for each order will not exceed the load capacity of the picking robot.
8. At the moment, the division of order orders is not contemplated. That is, only the batches of closed orders can be prepared.
9. The concept testing will be done on all AWS-Braket, Pennylane, D-Wave, and Qiskit environments. And we will stick with the scenario that best benefits our proof of concept.
10. We will use the docplex [70] to model our formulation.

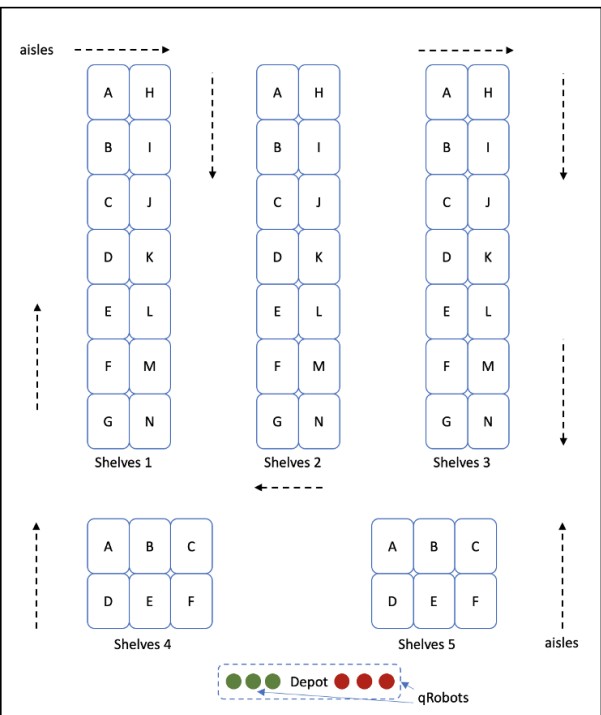

**Figure 7.** Structure of our warehouse with pick locations. The warehouse has a rectangular layout with no unused space. We use all the parallel corridors. This proof of concept contemplates a single warehouse used to take the order and deliver it, and it is also divided into blocks, which contain slots for storing products, and transverse aisles separate them. The cross aisles do not have any products but allow the collector to navigate in the warehouse. We base our picking strategy on minimizing the route and optimizing batch preparation. We do not contemplate shelving of different levels.

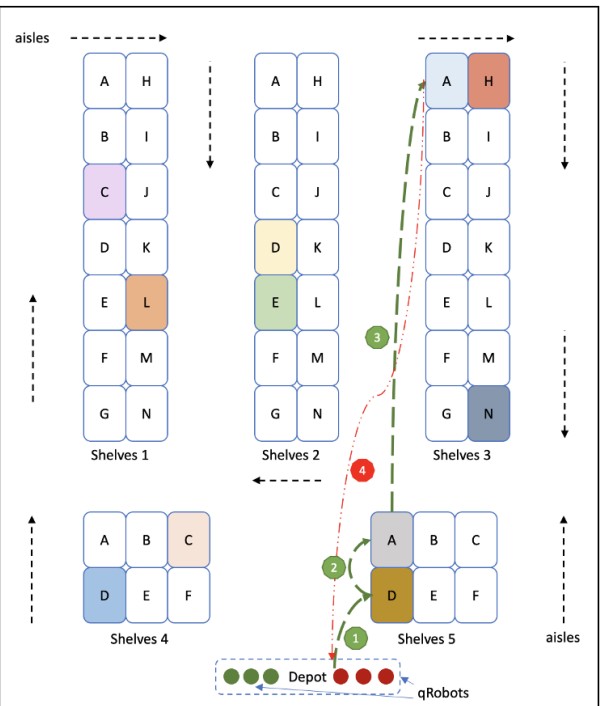

**Figure 8.** Scenario 1, Independent lots. The robot receives the orders and calculates which order is the most optimal according to the coordinates in which each product is found. In this example, lot 4 is the most optimal.

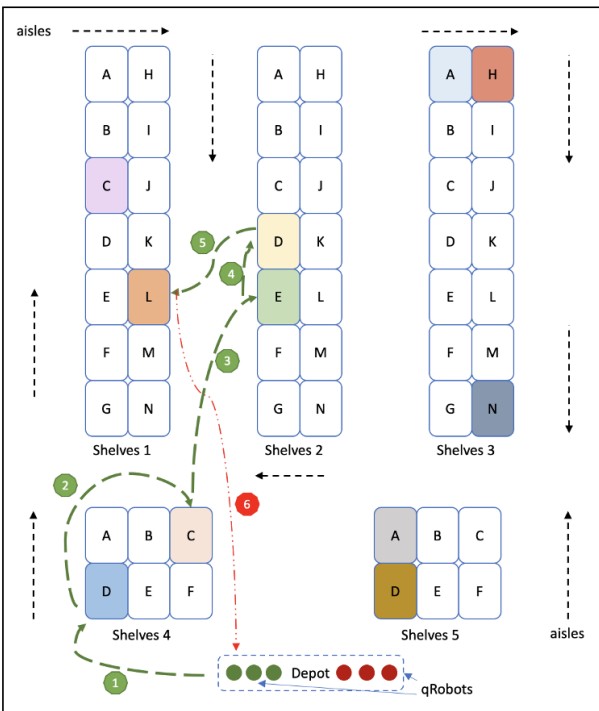

**Figure 9.** Scenario 2, Collecting products in the same route from different batches. The robot will calculate a path that includes all the products to optimize their collection in a single journey. For example, if the product from Lot 2 is next to one from Lot 1, the robot will pick it up and store it in the basket from Lot 2.

### 4.2. Picking and Batching Formulation

The formulation is represented as follows. In this scenario, the travel load is represented according to the number of robots we have. Let us imagine that we have several robots and that each of them makes a single trip. It would be the same as saying that we have a single robot that makes $n$ trips.

Let $N_0$ be the origin node, let $N_1 \ldots N_n$ be the nodes of the products, let $W_1 \ldots W_n$ be the weights in kg associated with for each product, let $d_{i,j}$ be the distance from node $i$ to $j$, let $M$ be the maximum load of the qRobots, let $K$ be the number of qRobots available, let $t$ be the instant, $i$ the node (product), $p$ the robot, and let $x_{t,i,p}$ be our binary variable (for example, for $x_{2,3,2} = 1$. It means that, at time 2, the qRobot 2 is at node 3). In our formulation, time really tells us the order, i.e., $t = 0$ will be the origin and $t = 1$ the moment in which it goes for the first batch. At $t = 2$ will be the moment of the second, and so on.

$$\min_{x} \quad \sum_{p=1}^{K} \sum_{t=1}^{n+1} \sum_{i=0}^{n} \sum_{j=0}^{n} x_{t_{t-1},i,p} x_{t,j,p} d_{i,j}, \tag{2}$$

s.t.

$$\sum_{p=1}^{K} x_{0,0,p} = K \tag{3}$$

$$\sum_{p=1}^{K} x_{n+1,0,p} = K, \tag{4}$$

$$\sum_{t=1}^{n+1} \sum_{i=1}^{n} x_{t,i,p} Wi \leq M \qquad \forall p \in \{1, ..., K\}, \tag{5}$$

$$\sum_{i=1}^{n} x_{t,i,p} = 1 \quad \forall t \in \{1, \ldots, n+1\}, \tag{6}$$

$$\forall p \in \{1, \ldots, K\}$$

$$\sum_{t=1}^{n+1} \sum_{p=1}^{K} x_{t,i,p} = 1 \quad \forall i \in \{1, \ldots, n+1\}, \tag{7}$$

$$x_{t,i,p} \in \{0,1\} \quad \forall t \in \{0, \ldots, n+1\}$$

$$\forall i \in \{0, \ldots, n\}. \tag{8}$$

$$\forall p \in \{1, \ldots, K\}$$

Equation (2) is our new objective function. Here, we minimize the total distance. We add the distance of all the robots traveling all the time, and we will check the distances of the nodes. Restriction (3) establishes that all the qRobots start from Depot. Restriction (4) establishes that all the qRobots end at Depot. Constraint (5) establishes any robot $p$ can carry more load than allowed. Constraint (6) declares that each robot can only be one node at any time. (7) establishes that throughout the entire route, the robots together pass each node only once, and Restriction (8) describes that $x_{t,i,p}$ are binary variables.

The number of the qubits to perform this algorithm is equal to $K(n+1)(n+2) + K\lceil log_2 M \rceil$. At this point, we can only map our objective function in quantum and then solve it with a VQE.

### 4.3. Mapping the Classical to Quantum Optimization

A common method for mapping classic optimization problems to quantum hardware is by coding it into the Hamiltonian [71] of an Ising model [58].

$$H_{Ising} = \sum_{i<j} J_{i,j} \sigma_i \sigma_j + \sum_i h_i \sigma_i, \tag{9}$$

where $\sigma_i$ is the product of $n$ identity matrices $I$ except a gate $Z$ in the $i$-th position and $\sigma_i \sigma_j$ product of identities minus gates $Z$ in positions $i$ and $j$.

As we already can build our objective function as a QUBO in the form $\langle x^T | Q | x \rangle$, now we can map our QUBO to Ising Hamiltonian formulation leads to calculating the values of $J_{ij}$ and $h_i$.

The transformation between QUBO and Ising Hamiltonian and is $z_i = 2x_i - 1$, where $z_i$ is a new variable that can take the values $-1$ or $1$. This means that by writing an algorithm for QUBO with this single variable change, we will have the algorithm in Ising form. That is very useful to have the algorithm for various platforms that are based on quantum gates (IBM Q and Pennylane) or quantum annealing (meanly D-Wave) in case of going from the Hamiltonian form. We can now solve our Picking and Batching Problem with VQE $\langle \psi(\theta) | H | \psi(\theta) \rangle$.

## 5. Results

Before analyzing in detail all the results of our proof of concept, it is of the utmost importance that we validate its operation globally and affirm that qRobot meets our expectations and works as we expected. Let us split the results of this proof of concept in two: 1, the configuration and conversion results of the Raspberry Pi 4 in a quantum computing environment (Figures 10–13); and 2, the picking and batching algorithm results represented by Tables 1–3 on one side and Figures 14 and 15 on the other.

The block diagram (Figure 16) summarizes the result of the implementation of the qRobot. The first thing we did is determine the mathematical model of our problem. We

then used the Docplex to model our objective function, along with its constraints. For our proof of concept, we used the Docplex library packages to move from Docplex to Qubo. From this point on, we had two possible operations according to our objectives. First, we modeled the problem for computers based on quantum gate technology, like IBMQ, and, second, for annealing computers, like D-Wave. Our experiments used both the Exact solver and VQE for tests on the Qiskit framework as samples based on quantum gates. But, before using the VQE, we needed to map the Qubo model to the Ising model. When we used the D-Wave computer, we only needed to reform the Qubo output list from Docplex to the Qubo format of the D-Wave computer.

```
pi@raspberrypi:~/qRobot/libcint/build $ python3
Python 3.7.3 (default, Jan 22 2021, 20:04:44)
[GCC 8.3.0] on linux
Type "help", "copyright", "credits" or "license" for more information.
>>> import tensorflow as tf
>>> tf.__version__
'2.3.1'
>>> import pennylane as qml
>>> qml.__version__
'0.14.1'
>>> import braket._sdk as braket_sdk
>>> braket_sdk.__version__
'1.5.15'
>>> import qiskit
>>> qiskit.__qiskit_version__
{'qiskit-terra': '0.17.1', 'qiskit-aer': None, 'qiskit-ignis': '0.6.0', 'qiskit-ibmq-provider': '0.12.2', 'qiskit-aqua
': None, 'qiskit': '0.25.1', 'qiskit-nature': '0.1.1', 'qiskit-finance': None, 'qiskit-optimization': '0.1.0', 'qiskit
-machine-learning': None}
>>> exit()
pi@raspberrypi:~/qRobot/libcint/build $ 
```

**Figure 10.** This figure shows that we judge important environments to carry out quantum computing to robotics and beyond. We can see the correct installation of TensorFlow 3.2.1 as required for all gradient operations; see the installation of Pennylane version 14.1, the installation of the latest version of Amazon Braket, and all the packages of the newest version of qiskit 0.25 minus the qiskit-machine-learning package.

```
[pi@raspberrypi:~/qRobot/libcint/build $ jupyter notebook --version
 6.3.0
[pi@raspberrypi:~/qRobot/libcint/build $ jupyter --version
 jupyter core     : 4.7.1
[jupyter-notebook : 6.3.0
 qtconsole        : 5.0.3
 ipython          : 7.22.0
 ipykernel        : 5.5.3
 jupyter client   : 6.2.0
 jupyter lab      : not installed
 nbconvert        : 5.4.1
 ipywidgets       : 7.6.3
 nbformat         : 5.1.3
 traitlets        : 5.0.5
pi@raspberrypi:~/qRobot/libcint/build $ 
```

**Figure 11.** In this figure, we can see the correct installation of the Jupyter package and the Jupyter notebook that has been our environment of proof of concept. With this, everything is ready to import or write code in the different frameworks mentioned above (IMBQ, AWS-Braket, Pennylane, and D-Wave).

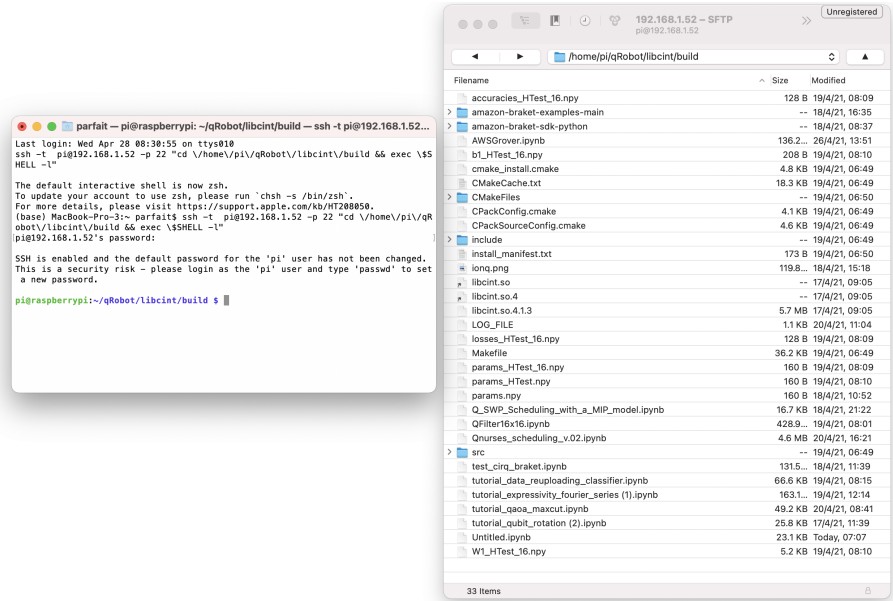

**Figure 12.** This figure shows the files window through the CyberDuck client SSH [72] viewer with the directory and file structure. On the left, you can see the terminal that gives access to the qRobot. To access the qRobot by SSH, the username and password are required. Everything is configurable [73].

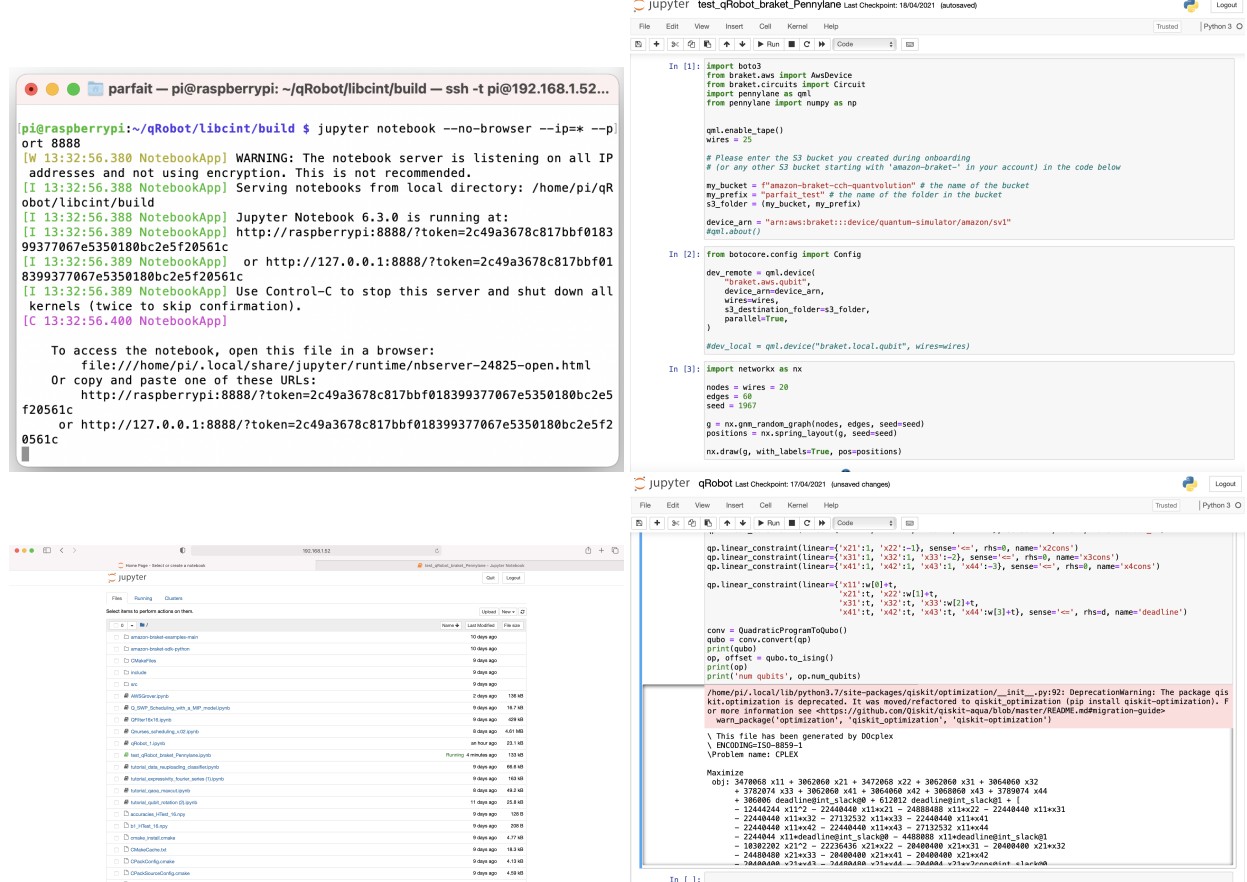

**Figure 13.** In these figures, we see several notebook tests in operation. Works with the Qiskit, Pennylane [74], and AWS-Braket frameworks' [75]. It was also tested with quantum computers, Rigetti [76], qiskit [69,77], and D-Wave [78]. In the figure of the terminal, you can observe the executions in progress. We can see from qiskit the docplex [70] in execution. From AWS and Pennylane [79,80], we can see how to call the quantum device from the Raspberry.

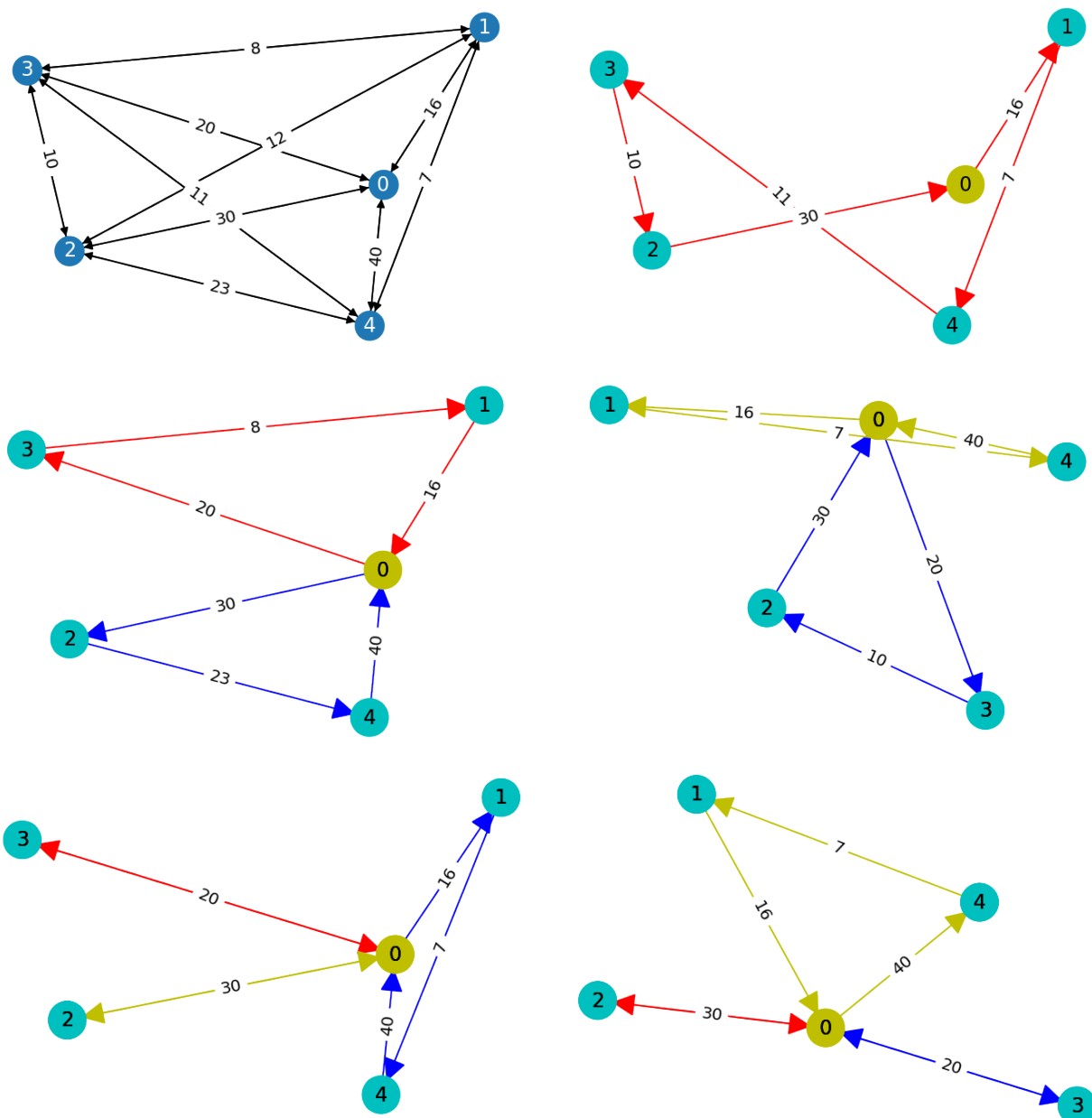

**Figure 14.** In these graphs, we can observe the results of the algorithm in different scenarios. A different color represents each qRobots; qRobot number 1 is red, the next is blue, the third is yellow, and so on, while the depot is the 0 node in yellow color, and the rest of the nodes are represented in blue. The weights of each item (not normalized) in kg are $w_0 = 0, w_1 = 8, w_2 = 8, w_3 = 3$, and $w_4 = 3$. The maximum capacity of each qRobots is 45. In this case, we have 4 items and the possibility of using up to 3 qRobots. Reading the images from left to right, we see that the nodes and their respective distances are shown in the first image. The second image shows the result of the algorithm having a qRobot. In the third and fourth images, we can see two different cases solved by two qRobots. And, finally, in the fifth and sixth images, we can see two other issues solved by three qRobots. It is important to highlight that our algorithm in this proof of concept minimizes the distance traveled and optimizes the number of qRobots necessary to solve the cases presented. If it judges that the task can be performed with a single qRobot, it will not send 2 qRobots.

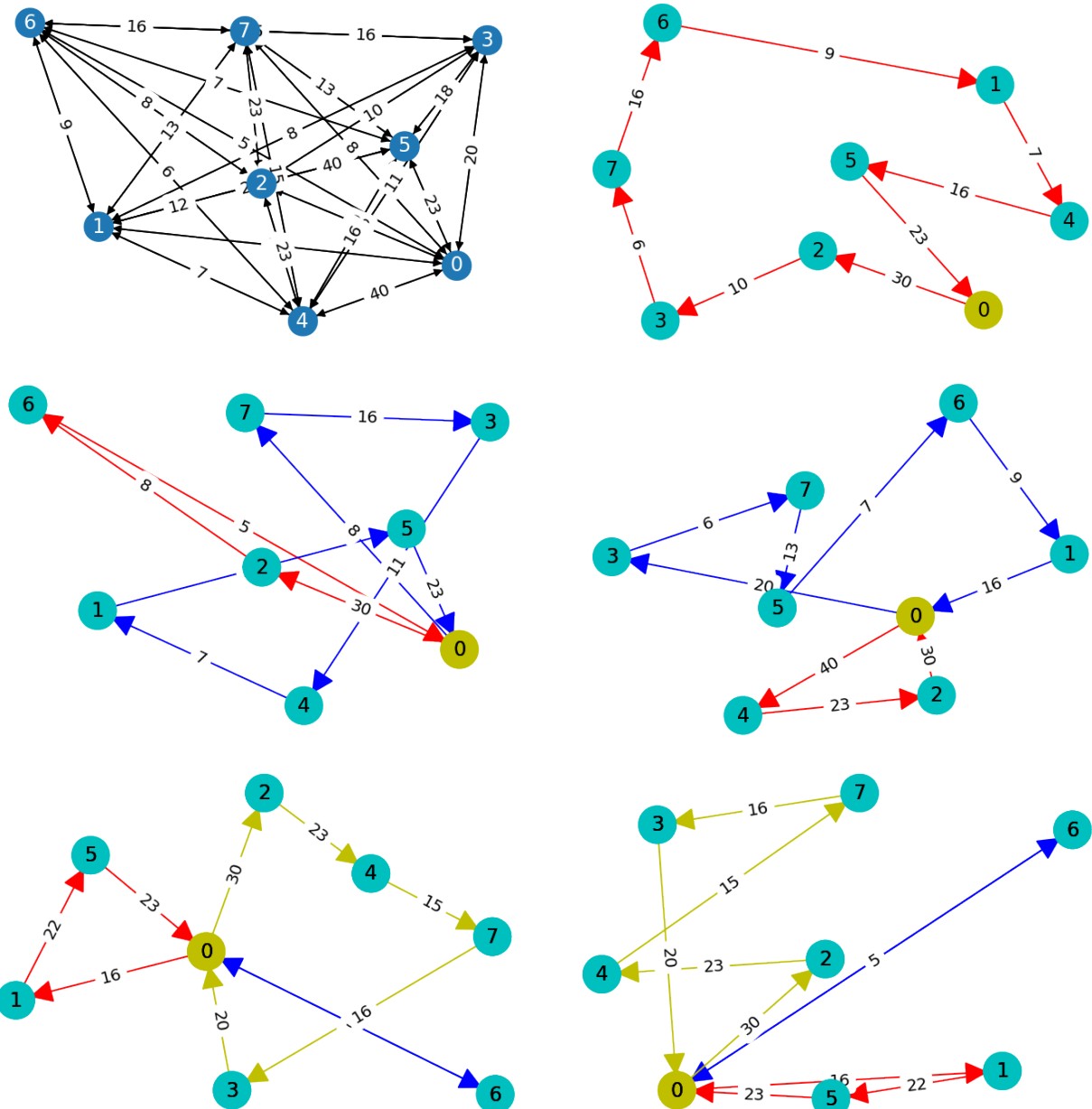

**Figure 15.** In these graphs, we can observe the results of the algorithm in different scenarios. A different color represents each qRobots; qRobot number 1 is red, the next is blue, the third is yellow, and so on, while the depot is the 0 node in yellow color, and the rest of the nodes are represented in blue. The weights of each item (not normalized) in kg are $w_0 = 0, w_1 = 8, w_2 = 8, w_3 = 3, w_4 = 3, w_5 = 1, w_6 = 2$, and $w_7 = 4$. The maximum capacity of each qRobots is 45. In this case, we have 7 items and the possibility of using up to 4 qRobots. Reading the images from left to right, we see that the nodes and their respective distances are shown in the first image. The second image shows the result of the algorithm having a qRobot. In the third and fourth images, we can see two different cases solved by two qRobots. And, finally, in the fifth and sixth images, we can see two other issues solved by three qRobots. It is important to highlight that our algorithm in this proof of concept minimizes the distance traveled and optimizes the number of qRobots necessary to solve the cases presented. If it judges that the task can be performed with a single qRobot, it will not send 2 qRobots.

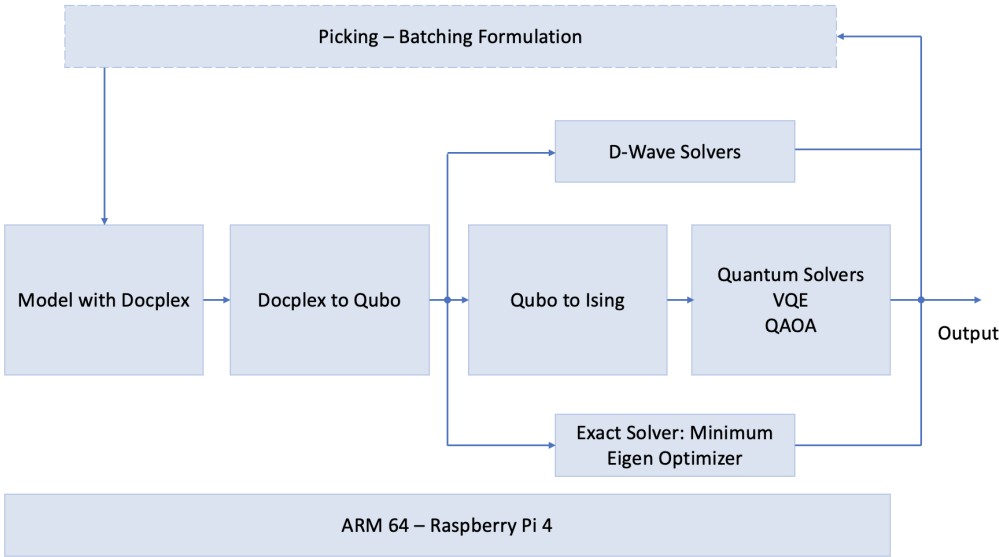

**Figure 16.** qRobot operation diagram. This diagram shows all the necessary blocks and processes that allow modeling the picking-batching problem and its proper functioning on the Raspberry Pi 4.

The steps to convert the Raspberry Pi 4 into a "quantum computer" are in the Appendix A.

Table 1 shows the experimentation results by setting the number of qRobots as their capacities (maximum load) at 1 and 45, respectively. We compared the execution time of our algorithm with different public access simulators on the market during this experimentation, solving the problem of picking and batching. We observed that, for issues of this nature, and especially due to the number of qubits required in each scenario, the behavior of the D-Wave is the desired one at the temporal level, comparing it with Gate-based Quantum Computing. However, it should be taken into account that, for experiments with numbers of qubits less than 20, the behavior of these simulators is equated with the D-Wave. This experimentation helps to have a clear vision about the feasibility of this proof of concept.

Continuing with the analysis of the results, Table 2 shows us the computational results of our picking and batching algorithm considering 1 qRobot through AWS-Braket and on the real quantum computer D-Wave Advantage_system1 [81]. The time value is an average and not counting latency time, job creation, and job return time.

We also analyze the latency time when running the algorithm from the qRobot to the quantum computer. The quantum computer was in Oregon (U.S.) and our qRobot in Barcelona (Spain) and Segovia (Spain) in the tests we had done. Out of all the tests we have run, we had an average latency time of around 2 s plus all job management processes rising to roughly 8 s. For the number of qubits greater than 30, it is very convenient to use AWS-Braket (Advantage_system1.1) instead of Qiskit or Pennylane for the number of qubits and the execution time; it is differentially better. This scenario makes the use of quantum in robots very viable. For tests with a value of $M$ less than those in the table, the number of qubits is relaxed, and the execution time is improved. This leads us to normalize the weights of the batches since the number of qubits follows the formula $K(n+1)(n+2) + K\lceil log_2 M \rceil$, where the $K\lceil log_2 M \rceil$ qubits are needed as ancillaries qubits.

We also analyze the quantum real-time execution deeply through Table 3. We have measured the execution time without counting the latency time, creating jobs, and returning the work.

Figure 14 offers us the algorithm results in different scenarios where we analyze some important cases, which helped us determine viable strategies within our proof of concept. It is important to note that our algorithm minimizes the distance traveled and optimizes

the number of qRobots. Figure 15 repeats almost the same scenario but now considering 7 items with the same number of qRobots.

**Table 1.** In this experimentation, both the number of qRobots and their capacities (maximum load) are fixed and are worth 1 and 45, respectively. We compare the execution time of our algorithm in the different public access simulators in the market, solving the picking and batching problem. We see that, for issues of this nature, and especially for the number of qubits required in each scenario, the behavior of the D-Wave is the desired one at the temporal level, comparing it with technologies based on quantum gates. However, it should be noted that, for the experiments on numbers of qubits less than 20, the behavior of these simulators is equated with the D-Wave. This experimentation helps to have a clear vision about the feasibility of this proof of concept.

| The Benchmark of the qRobot's Algorithm in Different Quantum Simulators. | | | | |
|:---:|:---:|:---:|:---:|:---:|
| # of Items | # Qubits | DWave—Time (s) | Ibmq_qasm_simulator—Time (s) | Pennylane—Time (s) |
| 2 | 18 | 1.92 | 1.89 | 1.94 |
| 3 | 26 | 3.2 | 737.46 | 656.93 |
| 4 | 36 | 4.88 | - | - |
| 5 | 48 | 7.60 | - | - |
| 6 | 62 | 11.16 | - | - |
| 7 | 78 | 15.89 | - | - |
| 8 | 96 | 21.72 | - | - |
| 9 | 116 | 30.18 | - | - |
| 10 | 138 | 43.29 | - | - |
| 11 | 162 | 53.28 | - | - |
| 12 | 188 | 63.45 | - | - |

**Table 2.** The computational results of our picking and batching algorithm on only 1 qRobot. The value of time is an average and includes the waiting time, queue, execution, and return of the solution. In the case of K is equal to 2 for 9 items with the qRobot capacity equal to 15, the number of qubits is 188. The execution time is on AWS Braket and on the D-Wave Advantage_system1 quantum computer. We can realize that there is a latency time in executing the algorithm from the qRobot to the real quantum computer. In the tests we have done, the quantum computer is in the U.S. West (Oregon). Of all the tests that we have done, we have had an average latency time of about 2 plus all the work management processes that rises more or less to about 8 s. For the number of qubits exceeding 30, it is very convenient to use AWS-Braket (Advantage_system1.1) [81] instead of Qiskit or Pennylane. By the number of qubits and the execution time, that is differentially better. This scenario makes the use of quantum in robotics very viable. For the tests with a value of $M$ lower than those in the table, the number of qubits is relaxed, and the execution time is improved. This leads us to normalize the weights of the batches since the number of qubits follows the formula $K(n+1)(n+2) + K\lceil log_2 M \rceil$.

| # of Items | qRobot's Capacity | # Qubits | AWS-Braket [75] Average Time (s) | ibmq_qasm_simulator [69] Average Time (s) | Pennylane [74] Average Time (s) |
|:---:|:---:|:---:|:---:|:---:|:---:|
| 2 | 15 | 10 | 11.23 | 0.053 | 0.041 |
| 3 | 15 | 16 | 22.96 | 0.40 | 0.27 |
| 4 | 15 | 24 | 33.07 | 537.46 | 480 |
| 5 | 15 | 34 | 57.93 | — | — |
| 6 | 15 | 46 | 118.41 | — | — |
| 7 | 15 | 60 | 145.83 | — | — |
| 8 | 15 | 76 | 296.81 | — | — |
| 9 | 15 | 94 | 335.64 | — | — |
| 10 | 25 | 115 | 427.36 | — | — |
| 11 | 25 | 137 | 650.25 | — | — |
| 12 | 25 | 161 | 908.71 | — | — |

**Table 3.** In this table, we only consider the running time of the quantum algorithm on the real quantum computer from the qRobot (Advantage_system1.1 [81]), not counting latency time, job creation, and job return time.

| # of Items | qRobot's Capacity | # Qubits | AWS-Braket [75] Average Time (s) | ibmq_qasm_simulator [69] Average Time (s) | Pennylane [74] Average Time (s) |
|---|---|---|---|---|---|
| 2 | 15 | 10 | 0.13 | 0.053 | 0.041 |
| 3 | 15 | 16 | 0.31 | 0.40 | 0.27 |
| 4 | 15 | 24 | 1.69 | 537.46 | 480 |
| 5 | 15 | 34 | 7.93 | – | – |
| 6 | 15 | 46 | 11.31 | – | – |
| 7 | 15 | 60 | 22.30 | – | – |
| 8 | 15 | 76 | 36.11 | – | – |
| 9 | 15 | 94 | 54.01 | – | – |
| 10 | 25 | 115 | 80.40 | – | – |
| 11 | 25 | 137 | 139.67 | – | – |
| 12 | 25 | 161 | 195.60 | – | – |

## 6. Discussion

We have achieved that, given a warehouse with a single robot, a list of several products with their respective loads and a list of batches, our system minimizes the distance to collect all the products and prepare the batches. This formulation solves the order in which the robot could manage all the products and make the batches passing through the depot. Another important achievement that this approach offers is that each robot makes a single trip. However, it is possible to band the code so that, if we find ourselves in a situation where there are many batches to create and only a few robots to do the picking, these robots can be made to make the necessary trips if we have $k$ qRobots that make, at most, one trip (we will never need more with $n$ batches). In this way, we will obtain all the packages for trips that we are interested in doing. A more understandable way of explaining it would be that when the first qRobot has finished its journey, it should only be ordered to do the one that would have made the qRobot $k + 1$, which does not exist and so on with all the qRobots $k + 2$, $k + 3$, $k + 4$ ... until all scheduled batches are finished.

Right now, in addition to the processor, quantum computing simulation is closely related to memory. What takes up memory is to simulate a quantum computer, but the quantum computer does not need that memory, so it is assumed that it will end up being better. In this proof of concept, using 8 GB of RAM on the Raspberry Pi 4, we got the following results. The algorithm of collection and generation of packages take between 2 and 450 s to generate the batches and picking. If you want the qRobot to do all these tasks, we need to calculate the path before forming the packs. That said, we must bear in mind that, if what we want is to recalculate new routes when the robot has already left, we must take into account a lower latency time but close to said interval. A possible solution would be to choose a Raspberry with more RAM capacity. For example, if we had a 64 GB Raspberry Pi, this time would be cut to 2/8, and it would take approximately 56.25 s (less than a minute) to create the batches. However, in this era of quantum computing, it is not representative to compare times since the computational differences will be noticed when the problems begin to grow, not on the small scales that we are currently dealing with.

Effective viability for today's warehouses would consist of splitting the tasks of the robots and having a qRobot that centralizes all the requests and passes them to the fleet of $n$ qRobots so that they collect the products belonging to each batch.

We also performed tests and developed a system that allows us to model the problem and run it on a D-Wave. Despite the optimization of the algorithm, the number of necessary qubits ($K(n+1)(n+2) + K\lceil log_2 M\rceil$) and the need for low latency make this code adapted to the Annealing model. For this reason, we have prepared the Raspberry PI so that it can run D-Wave directly and under Amazon-braket-ocean-plugin. For more information, see

the steps in Appendix A. With this scenario, one could have a "reasonable" latency for low data volume, things that, today, computers based on quantum gates cannot offer.

## 7. Conclusions and Further Work

As we have seen, the problem raised throughout this work offers us an efficient way of managing a series of $K$ qRobots to collect a set of orders, optimizing the number of robots used. The provided approach applies to a "central computer" capable of carrying out all the calculations and then giving each of the robots' orders. However, when we begin to deal with very large problems both in the number of products and in the number of robots, the number of qubits required will tend to grow too large. A possible solution is to distribute the calculation of a central computer to each of the robots in such a way that each one has to calculate its route given a list of products to be collected. In this case, the equations of the problem would not change; just take $K = 1$ for each qRobot and apply the technique mentioned at the beginning of the discussion. Although it may not be possible to reach the best solutions, this process of distribution of the calculation would suppose a significant computational cost reduction despite the need to create the batches beforehand. This search for batch creation will be studied in future projects. On the other hand, it is important to note that the problem dealt with has a QUBO-type formulation, which allows it to be executed in annealing-type quantum computers. This makes a big difference in today's era (NISQ) as we have managed to work with 200 qubits versus the 30 qubits that we would have with a gate-based quantum computer. Finally, note that the defined problem seeks to minimize the total distance traveled by the robots, making it worthwhile for not all the robots to come out. For future lines, we will address the same problem. Still, We will continue to try to reduce the total times instead of the distance traveled (as done in this previous work [82]) since this situation is also very important in warehouse logistics.

**Author Contributions:** Conceptualization, P.A.-A.; Methodology, P.A.-A. and G.A.-L. and J.A.-C.; Software, P.A.-A. and G.A.-L.; Validation, P.A.-A., G.A.-L., J.A.-C. and D.C.-F.; Formal Analysis, P.A.-A., G.A.-L. and J.A.-C.; Investigation, P.A.-A.; Resources, P.A.-A.; Data Curation, P.A.-A.; Writing–Original Draft Preparation, P.A.-A.; Writing–Review and Editing, P.A.-A., J.A.-C. and G.A.-L.; Visualization, P.A.-A., G.A.-L., J.A.-C. and D.C.-F.; Supervision, P.A.-A.; Project Administration, J.A.-C. All authors have read and agreed to the published version of the manuscript.

**Funding:** This research received no external funding.

**Institutional Review Board Statement:** This article does not contain any studies with human or animal subjects.

**Informed Consent Statement:** Informed consent was obtained from all individual participants included in the study.

**Data Availability Statement:** Data sharing not applicable. No new data were created or analyzed in this study. Data sharing is not applicable to this article

**Acknowledgments:** The authors greatly thank the AWS-Braket and IBM team, mainly Simone Severini and Steve Wood, respectively. P.A. thanks Jennifer Ramírez Molino for her support and comments on the manuscript.

**Conflicts of Interest:** The authors declare no conflict of interest.

## Appendix A. Installation of ARM64 on Raspberry Pi 4

This section will describe step by step and delve into how installing and running Pennylane, AWS-Braket, D-Wave-Ocean, Qiskit, on a Raspberry Pi 4 under the ARM64 [68] operating system torn it into a quantum computing simulator and use it to access real quantum computers from IBMQ [69,77], AWS-Braket [75], D-Wave [78], and Regetti [76]. These frameworks and packages are required for the proof of concept that we propose.

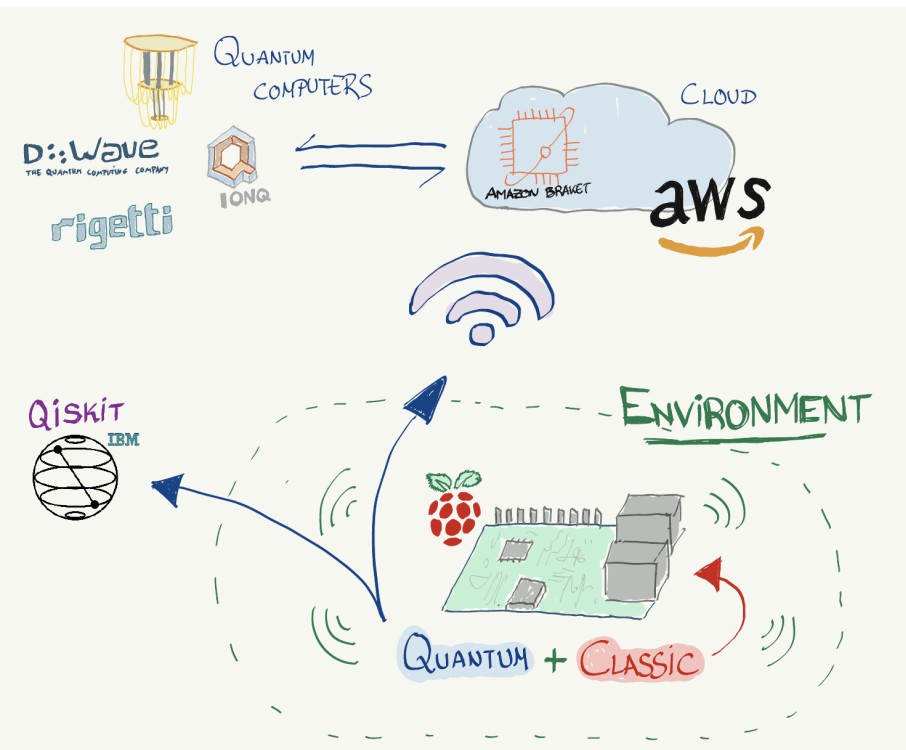

**Figure A1.** We have installed the following frameworks successfully (Qiskit, Pennylane, AWS-Braket) on our Raspberry Pi 4 under the ARM64 operating system. More information about the qRobot platform can be found at Ref. [83].

## Listing A1: Installation Raspberry Pi 4 ARM64

```
1  Steps
2  1. Download the latest image of Raspberry Pi ARM64
3  2. Initial setup of a headless Raspberry Pi
4  3. Setup of the Python environment and TensorFlow 2.3.1
5  4. Manual installation of some dependencies
6  5. Installation of the Qiskit elements
7  6. Installation of the Pennylane elements
8  7. Installation of the Amazon elements
9  8. Setup of Jupyter Notebooks
10 9. Enable remote desktop access using VNC
11 10. Test Jupyter notebook codes
12 11. Install DWave framework and Amazon-braket-ocean-plugin
13
14 Note: The actual version of the ARM64 for Raspberry Pi 4 is
       not stable. https://www.raspberrypi.org/forums/viewtopic.php?t=275370
15
16 1. Download the latest image of Raspberry Pi ARM64
17 Download the image from: https://downloads.raspberrypi.org/raspios_arm64/
       images/raspios_arm64-2021-04-09/
18
19 2. Initial Setup of a headless Raspberry Pi
20 We want to setup a headless Raspberry Pi (i.e. without display, keyboard,
       mouse), and also
       not use display/keyboard/mouse during the setup procedure.
21 Creating an SD card with the initial OS is described at https://www.
       raspberrypi.org/documentation/installation/sdxc_formatting.md
22 We will use the Raspberry Pi Imager and choose %Raspberry Pi OS Desktop (64-
       bit)% to write the image to the SD card. It
       is recommended to use the Desktop image vs. the other alternatives.
23
24 Prepare for wireless boot
25 According to https://www.raspberrypi.org/documentation/configuration/wireless
       /headless.md, we create a file wpa_supplicant.conf
```

```
       in the root directory of the SD card with the following content (replace
          (DE) with the appropriate country code, and (SSID)
          and (WLAN PASSWORD) with the SSID and password for our WLAN
          access point):
26 country=DE
27 ctrl_interface=DIR=/var/run/wpa_supplicant GROUP=netdev
28 update_config=1
29 network={
30    ssid="SSID"
31    psk="WLAN PASSWORD"
32 }
33 Now, we boot the Raspberry Pi, i.e., insert the SD card into the Raspberry Pi

          and connect it to a power supply.

34
35 3. Set up the Python environment
36 Do not use conda/anaconda/berryconda as recommended on other hardware
          platforms
          for Qiskit you can use the new virtual environment as you judge it
          suitable.

37
38 TensorFlow 2.3.1 for Python 3.
39 The whole shortcut procedure is found below. The wheel was too large to store
           at GitHub, so Google drive
          is used. Please make sure you have the latest pip3
          and python3 version installed; otherwise, pip may come with the message "
          .whl is not a supported wheel on this platform".

40
41 Check your Python3 version. Each version needs a unique wheel. Currently, the
           Raspberry Pi 64-bit operating system uses Python 3.7.3. So, you need to
          download Tensorflow-2.3.1-cp37-cp37m-linux_aarch64.whl. Undoubtedly, the
          Python version will upgrade over time
          and you will need a different wheel. See out GitHub page for
          all the wheels.
42 # get a fresh start (remember, the 64-bit OS is still under development)
43 $ sudo apt-get update
44 $ sudo apt-get upgrade
45 # install pip and pip3
46 $ sudo apt-get install python-pip python3-pip
47 # remove old versions, if not placed in a virtual environment (let pip search
          for them)
48 $ sudo pip uninstall tensorflow
49 $ sudo pip3 uninstall tensorflow
50 # install the dependencies (if not already onboard)
51 $ sudo apt-get install gfortran
52 $ sudo apt-get install libhdf5-dev libc-ares-dev libeigen3-dev
53 $ sudo apt-get install libatlas-base-dev libopenblas-dev libblas-dev
54 $ sudo apt-get install liblapack-dev
55 # upgrade setuptools 47.1.1 -> 50.3.0
56 $ sudo -H pip3 install --upgrade setuptools
57 $ sudo -H pip3 install pybind11
58 $ sudo -H pip3 install Cython==0.29.21
59 # install h5py with Cython version 0.29.21 (6 min @1950 MHz)
60 $ sudo -H pip3 install h5py==2.10.0
61 # install gdown to download from Google drive
62 $ pip3 install gdown
63 # copy binairy
64 $ sudo cp ~/.local/bin/gdown/usr/local/bin/gdown
65 # download the wheel
66 $ gdown https://drive.google.com/uc?id=1jbkp2rSZZ3YY-AM1vuHyB9hI05zrZGHg
67 # install TensorFlow (63 min @1950 MHz)
68 $ sudo -H pip3 install tensorflow-2.3.1-cp37-cp37m-linux_aarch64.whl

69
70 When the installation
          is successful, you should get the following screendump by executing:
71 $ python3
72 >>> import tensorflow as tf
73 >>> tf.__version__
74 you may have 2.3.1
```

```
75
76 Now you may install the pyscf
       for more information: http://pyscf.org/pyscf/install.html#compiling-from-
       source-code
77
78 Prerequisites for manual install are
79 *    CMake >= 3.10
80 *    Python >= 3.6
81 *    Numpy >= 1.13
82 *    Scipy >= 0.19
83 *    h5py >= 2.7
84 You can download the latest version of PySCF (or the development branch)
       from GitHub:
85 $ git clone https://github.com/pyscf/pyscf.git
86 $ cd pyscf
87 $ git checkout dev # optional if you would like to try out the development
       branch
88 Next, you need to build the C extensions in pyscf/lib:
89 $ cd pyscf/lib
90 $ mkdir build
91 $ cd build
92 $ cmake ..
93 $ make #(30 min)
94
95 export PYTHONPATH=/opt/pyscf:$PYTHONPATH
96
97 please check if the package hs been installed successfully
98 >>> import pyscf
99
100 Execute this:
101 cd pyscf/lib
102 sh _runme_to_fix_dylib_osx10.11.sh
103
104
105 4. Manual installation of some dependencies
106  Based on and taking advantage of @Jan Lahmann, we need to install
       and configure some prerequisites first manually.
107 retworkx
108 We will install retworkx according to the instructions
       in https://retworkx.readthedocs.io/en/stable/README.html#installing-
       retworkx. First, install the rust language environment.
109 pi$ cd ~/qrobot
110 pip install setuptools-rust
111 curl -o get_rustup.sh -s https://sh.rustup.rs
112 sh ./get_rustup.sh -y
113 Now activate rust and install retworkx:
114 pi$ source ~/.cargo/env
115 pip3 install retworkx
116
117
118 5. Installation of the Qiskit elements
119 After the pre-work we just completed, installing Qiskit should now be as
       simple as
120
121 pip3 install --force-reinstall pip
122 #pip3 install vaex
123 sudo apt install llvm-7-dev
124
125 #I recommend to install separely each paquet from qiskit. The version of the
       installed qiskit is 0.25.1
126 #In this version, you will not be able to install qiskit-machine-learning
127 pip3 install qiskit-aqua
128 pip3 install qiskit-aer
129 pip3 install 'qiskit[visualization]'
130
131 #Now, let us see what versions of Qiskit were installed:
132 pip3 list | grep qiskit
133 qiskit 0.25.1
134 qiskit-aer 0.8.1
```

```
135  qiskit-aqua 0.9.1
136  qiskit-finance 0.1.0
137  qiskit-ibmq-provider 0.12.2
138  qiskit-ignis 0.6.0
139  qiskit-nature 0.1.1
140  qiskit-optimization 0.1.0
141  qiskit-terra 0.17.1
142
143  python --version
144  >>>Python 3.7.3
145
146  Command "python setup.py egg_info" failed with error code 1
         in /tmp/pip-install-eur2lck3/qiskit-aer/
147
148  6. Installation of the Pennylane elements
149  pip install pennylane --upgrade
150  pip install autograd
151
152  7. Installation of the Amazon elements
153  pip install amazon-braket-sdk
154  pip install amazon-braket-pennylane-plugin
155
156  Needing to set if you specify directly with boto3, it would be like this, but
          you are using PennyLane
157  https://boto3.amazonaws.com/v1/documentation/api/latest/guide/configuration.
         html
158  https://boto3.amazonaws.com/v1/documentation/api/latest/guide/configuration.
         html#using-a-configuration-
         file
159  aws_access_key_id and aws_secret_access_key will also be required, which are
         associated with AWS IAM User.
160
161  For that, you must need any ~/.aws/config file
162  Edit with:
163  cat ~/.aws/config
164
165  mkdir ~/.aws
166  touch ~/.aws/config
167  echo "[default]" >> ~/.aws/config
168  echo "region = us-east-1" >> ~/.aws/config
169  echo "aws_access_key_id = AKIAIOSFODNN7EXAMPLE" >> ~/.aws/config
170  echo "aws_secret_access_key = wJalrXUtnFEMI/K7MDENG/bPxRfiCYEXAMPLEKEY
         " >> ~/.aws/config
171
172  I use by default us-east-1, but the user can use the region he has.
173
174  the output format is like this example:
175  [default]
176  region = us-east-1
177  aws_access_key_id = AKIAIOSFODNN7EXAMPLE
178  aws_secret_access_key = wJalrXUtnFEMI/K7MDENG/bPxRfiCYEXAMPLEKEY
179
180  From here, you can be able to execute any code in Pennylane or AWS-Braket.
         For another platform, like D-Wave, you may need to install:
181  pip3 install amazon-braket-ocean-plugin
182  pip3 install dwave_networkx
183  pip3 install minorminer
184  pip3 install dwave-ocean-sdk
185
186  8. Setup of Jupyter Notebooks
187  pip3 install jupyter
188
189  Start Jupyter without a local browser and listen on port 8888
         for a remote connect:
190  jupyter notebook --no-browser --ip=* --port 8888
191
192  Please follow the message from your prompt. It might be necessary to replace
         the hostname (raspberrypi) with the correct hostname
         in our local network or the IP address of the raspberry.
```

```
193
194 You can configure to access as local and access the Jupyter notebook
        interface using the URL http://raspberrypi:8888/
        from a browser on our laptop. You will need to replace (raspberry) with
        the correct hostname
        or IP of the Raspberry Pi. For that, you need to execute the
        next command:
195
196 mkdir -p ~/qRobot/temp; cd ~/qRobot/temp;
197 jupyter notebook --no-browser
198
199 9. Enable remote desktop access using VNC
200 In addition to connecting to the Raspberry Pi via ssh, it might be useful to
        enable access with VNC to connect to a graphical desktop that
        is running locally on the Raspberry Pi. This is described at https://
        desertbot.io/blog/headless-raspberry-pi-4-remote-desktop-vnc-setup.
201 First, we enable VNC and change the screen resolution:
202  sudo raspi-config
203 Select Interfacing Options
204 Select VNC
205 For the prompt to enable VNC, select Yes (Y)
206 For the confirmation, select Ok
207 Select Advanced Options
208 Select Resolution
209 Select anything but the default (example: 1280x720)
210 Select Ok
211 Select Finish, Yes to reboot
212
213 For my local ssh viewer, I used Cyberduck https://cyberduck.io/, but you can
        also use https://www.realvnc.com/en/connect/download/viewer/
        and connect to the Raspberry Pi (enter the IP address
        in VNC viewer; enter login information). After the first connect, we will
         be asked to adjust some configurations (location settings, display
        settings, system update, etc.).
214
215 10. Test Jupyter notebook codes,
216
217 11. Install DWave framework and Amazon-braket-ocean-plugin
218 As DWave, unfortunately, does
        not provide ARM wheels yet. That means that you need to build
        from the source distributions. Simultaneously, dimod requires boost (
        https://github.com/dwavesystems/dimod#installation), though they are
        planning to remove that dependency soon (https://github.com/dwavesystems/
        dimod/issues/618, https://github.com/dwavesystems/dimod/pull/748).
219 You can try installing boost (https://www.boost.org/)
        and then trying to install dimod again.
220 The simple way
        is by installing as apt-get install libboost-dev. So, follow the
        next steps below. During these steps, you may need to upgrade your pip
        or NumPy.
221 apt-get install libboost-dev
222 pip3 install amazon-braket-ocean-plugin
223
224 After these steps, you must need to install the package from D-Wave.
225
226 By installing dwave-tabu from source on master (we switched
        from swig to cython, but haven't released 0.4 yet):
227 pip install -U pip setuptools
228 USE_CYTHON=1 pip install -e git+https://github.com/dwavesystems/dwave-tabu.
        git#egg=dwave-tabu
229
230 after this, install pip3 install dwave-system
231 Then, you already have your system ready to use DWave from your Raspberry PI
        4.
```

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
