# Peer review of "qRobot: A Quantum Computing Approach in Mobile Robot Order Picking and Batching Problem Solver Optimization"

_algorithms, doi:10.3390/a14070194_

Round 1
Reviewer 1 Report
This paper analyzes the simulation and implementation of quantum algorithms that find optimal solutions for picking and batching products stored in a warehouse. Before addressing the technical part, the authors provide an interesting review of papers on quantum algorithms for optimization problems. They also have provided a brief review of VQE. In the technical part, the authors map their optimization problem into the Ising model, which can be implemented in Dwave quantum computers. There is a comparison with IBMQ and Pennylane cross-platform implementations for the cases that use a small number of qubits (<30). I do consider this paper worthy of being published in Algorithms, but it is desirable that the authors improve some figures (the letters are too small in some of them), and be more fastidious regarding the references in the review part (for instance, give a reference instead of saying "the researcher Lucas". Cite [55] using the name of the first author together with et al instead of the second author. There seems to have problems with refs. [45], [56], [57].) Is listing 1 really necessary?
Author Response
>> Issue resolved. We have removed the author’s name and We reworded the sentence by setting the reference.
There seems to have problems with refs. [45], [56], [57].) Is listing 1 really necessary?
>> Issue resolved. We have removed reference 45 and 56 and 57 pointed to another reference.

Reviewer 2 Report
In this paper, authors hope to optimize the actions of robotics by using quantum variablal computing. They take use of a Raspberry Pi 4, generating a quantum combinatorial optimization algorithm that saves the distance travelled and the batch of orders. The simulations show that the present algorithms may be improved in NIST era. The result is interesting and may be valuable for publication.
(1) The transfer from classical combinatiorial optimization algorithm to quantum variablal problem is unclear. It may better to present an algorithm for explaining this procedure. And then, one simple example may be presented for readers.
(2) Although there are some simulations introduced, however, it is not very unclear, especially for readers who have not too much robots. It may more involved to present schematic procedure. For example, "The steps to convert the Raspberry Pi 4 into a "quantum computer" are in the Appendix A" this sentences and the followed appendix have not introduced the procedure schematially.
Hence, the main concern for me is to explain clearly how to transfer the quantum computation into robots.
Author Response
In this paper, authors hope to optimize the actions of robotics by using quantum variablal computing. They take use of a Raspberry Pi 4, generating a quantum combinatorial optimization algorithm that saves the distance travelled and the batch of orders. The simulations show that the present algorithms may be improved in NIST era. The result is interesting and may be valuable for publication.
(1) The transfer from classical combinatiorial optimization algorithm to quantum variablal problem is unclear. It may better to present an algorithm for explaining this procedure. And then, one simple example may be presented for readers.
(2) Although there are some simulations introduced, however, it is not very unclear, especially for readers who have not too much robots. It may more involved to present schematic procedure. For example, "The steps to convert the Raspberry Pi 4 into a "quantum computer" are in the Appendix A" this sentences and the followed appendix have not introduced the procedure schematially.
Hence, the main concern for me is to explain clearly how to transfer the quantum computation into robots.
>> In order to clarify all the points above we have added the following diagram and paragraph:
>> Fig(10) QRobot operation diagram. This diagram shows all the necessary blocks and processes that allow modelling the picking-batching problem and its proper functioning on the Raspberry Pi 4.
>> The block diagram (10) summarizes the result of the implementation of the qRobot. The first thing we did is determine the mathematical model of our problem. We then used the Docplex to model our objective function along with its constraints. For our proof of concept, we used the Docplex library packages to move from Docplex to Qubo. From this point on, we had two possible operations according to our objectives. First, we modelled the problem for computers based on quantum gate technology like IBMQ, and second, for annealing computers like D-Wave.
Our experiments used both the Exact solver and VQE for tests on the Qiskit framework as samples based on quantum gates. But before using the VQE, we needed to map the Qubo model to the Ising model. When we used the D-Wave computer, we only needed to reform the Qubo output list from Docplex to the Qubo format of the D-Wave computer.

Reviewer 3 Report
Thank you for submitting your interesting paper to the journal. I would like to add some of my comments to help you to improve your paper and increase its correctness and clarity.
Please find my comments specified according to the subsequent sections of the paper as follows:
- In the "Abstract", it would be good to briefly summarize the numerical results of the proposed approach.
- The "Introduction" needs to be modified. Please try to add few more references and justify your problem.
- The results should be thoroughly justified with respect to previous research papers.
- The figures should be on the same page as it is mentioned in the text. Figures 2 and 3 are not clear.
- Please consider the journal requirement for formatting the references.
- There are few grammatical errors. Please revise the text and proofread it.
Author Response
Reviewer 3
Thank you for submitting your interesting paper to the journal. I would like to add some of my comments to help you to improve your paper and increase its correctness and clarity.
Please find my comments specified according to the subsequent sections of the paper as follows:
- In the "Abstract", it would be good to briefly summarize the numerical results of the proposed approach.
ïƒ We added the summary of the results.
- The "Introduction" needs to be modified. Please try to add few more references and justify your problem.
ïƒ done.
- The results should be thoroughly justified with respect to previous research papers.
ïƒ As far as the authors know, this is a novel research without similar implementations. We have added references of research that pointed the problems to solve.
- The figures should be on the same page as it is mentioned in the text. Figures 2 and 3 are not clear.
ïƒ We tried to improve the location of the figures, but there is a trade-off between the paragraphs and the size of the image. We tried to add a reference that increases the clarity to follow the explanation and its connection to the images.
- Please consider the journal requirement for formatting the references.
ïƒ We tried to use the right template. If it’s still incorrect, let us know.
- There are few grammatical errors. Please revise the text and proofread it.
ïƒ Reviewed by a native English speaker.

Round 2
Reviewer 3 Report
Thank you for improving the manuscript. If possible, please try to improve the clarity of Figures 2 and 3.